# Effective Reduction in Nuclear DNA Contamination Allows Sensitive Mitochondrial DNA Methylation Determination by LC-MS/MS

**DOI:** 10.3390/ijms26188864

**Published:** 2025-09-11

**Authors:** Lin Liang, Luis Alfonso González Molina, Pytrick G. Jellema, Martijn van Faassen, Laura T. A. Otten, Kevin P. Mennega, Ingrid H. Hof, D. A. Janneke Dijck-Brouwer, Amalia M. Dolga, Marianne G. Rots, Klary E. Niezen-Koning

**Affiliations:** 1Department of Pathology and Medical Biology, University Medical Center Groningen, University of Groningen, 9713 GZ Groningen, The Netherlands; l.liang@umcg.nl (L.L.); l.a.gonzalez.molina@umcg.nl (L.A.G.M.);; 2Department of Molecular Pharmacology, University Medical Center Groningen, University of Groningen, 9713 AV Groningen, The Netherlands; 3Department of Laboratory Medicine, University Medical Center Groningen, University of Groningen, 9713 GZ Groningen, The Netherlandsk.p.mennega@umcg.nl (K.P.M.);

**Keywords:** neurometabolic disorders, mitochondrial DNA methylation, TRIzol RNA phase, LC-MS/MS, nuclear DNA

## Abstract

Mitochondria are essential organelles for cellular energy production, playing a central role in driving metabolic processes and supporting critical intracellular functions. Neurometabolic disorders encompass a wide variety of conditions characterized by mitochondrial dysfunction. Owing to their bacterial ancestry, mitochondria possess an independent genome consisting of a circular DNA molecule (mtDNA), which has been reported to be subject to methylation. However, the technical challenges in the detection of mtDNA methylation have led to debates on its existence. One of the concerns is that the compactness of mtDNA can lead to suboptimal bisulfite conversion, thereby causing mtDNA methylation overestimation. To address this, liquid chromatography tandem mass spectrometry (LC-MS/MS) offers a bisulfite-independent readout; however, this method requires mtDNA samples devoid of nuclear DNA (nDNA) contamination. To diminish nDNA contamination, we isolated mtDNA from the TRIzol RNA phase. Importantly, pyrosequencing showed no significant difference in the methylation levels of mtDNA isolated from the TRIzol RNA phase compared to those from the TRIzol DNA phase, or isolated via total genomic DNA (gDNA). Across different human cell lines, LC-MS/MS detected significantly lower global methylation levels for DNA isolated from the TRIzol RNA phase than those from the TRIzol DNA or gDNA isolation. Moreover, using mtDNA isolated from the TRIzol RNA phase, LC-MS/MS validated the enhanced mtDNA methylation in HepG2 transgenic cell lines expressing mitochondrial-targeted DNA methyltransferases (means of 2.89% and 2.03% for MCviPI and MSssI transgenic cell lines, respectively), compared to two negative control cell lines (1.36 and 1.39%). When applying it to clinically relevant material, LC-MS/MS demonstrated a significantly lower global methylation level for platelet DNA isolated from the TRIzol RNA phase (mean of 1.98%) compared to gDNA isolations (mean of 4.32%). Similar findings were confirmed in mouse brain tissue, in which a significantly lower methylation level was detected in DNA isolated from the TRIzol RNA phase (1.79%) compared to that from gDNA isolation (5.12%). In conclusion, isolating mtDNA from the TRIzol RNA phase holds significant potential in future studies, particularly for the quantification of mtDNA global methylation by LC-MS/MS, a technique that is independent of bisulfite conversion and bioinformatic analysis.

## 1. Introduction

Neurometabolic disorders encompass a group of biochemical and pathological alterations affecting human metabolism and brain function, often associated with mitochondrial dysfunction [1]. These neurometabolic disorders display diverse characteristics across various stages of life and can affect multiple organ systems [2,3]. Clinically, neurometabolic disorders are highly heterogeneous, arising from pathogenic mutations in both nuclear DNA (nDNA) and mitochondrial DNA (mtDNA) (Appendix A). These DNA mutations impair mitochondrial function and disrupt metabolism, neurotransmitter synthesis, and overall cellular homeostasis, ultimately contributing to neurodegeneration [4]. Mitochondria contain independent genomes encoding essential proteins for the electron transport chain (ETC), which generates about 90% of cellular ATP through oxidative phosphorylation (OXPHOS) [5]. Mitochondrial dysfunction can lead to severe consequences in the brain, a highly energy-demanding organ, resulting in metabolic disorders and neurodegeneration [6]. Increasing evidence indicates that mtDNA methylation is involved in mitochondrial dysfunction and associated with the pathogenesis of various diseases [7,8], including neurodegenerative diseases [9], metabolic disorders [10,11], and cardiovascular diseases [12,13]. However, the existence of mtDNA methylation remains controversial. The primary concerns are related to the technical limitations of reliably and accurately quantifying the globally very low methylation level [8,14,15,16].

Bisulfite (pyro)sequencing is the gold standard for measuring DNA methylation. This process converts unmethylated cytosine (C) into uracil (U), eventually detected as thymine (T) after PCR amplification, while methylated cytosine remains unchanged. However, the compact structure of mtDNA may hinder access for bisulfite conversion. Therefore, the incomplete conversion of unmethylated cytosines can occur, potentially leading to an overestimation of methylation levels [14,16,17]. To circumvent this issue, mtDNA linearization before bisulfite conversion has indeed increased bisulfite accessibility and lowered the methylation readout [18,19]. Alternatively, bisulfite-independent techniques, including long-read sequencing methods such as nanopore, can be used to quantify mtDNA methylation [11,20,21]. Nevertheless, inconsistent data has been reported with sequencing-based approaches regarding the existence of mtDNA methylation. For instance, studies carried out by Lüth et al. found very low levels of CpG methylation frequency in mtDNA from blood and midbrain neurons [20]. Mposhi and coworkers also detected mtDNA CpG methylation at low levels in wild-type HepG2 and confirmed the increased methylation induced by mitochondria-targeted methyltransferases [10]. Other sequencing studies suggested no evidence to support the presence of mtDNA methylation [19,22]. These inconsistent findings led us to investigate another alternative method for mtDNA methylation measurement.

Liquid chromatography tandem mass spectrometry (LC-MS/MS) is another bisulfite-independent method to quantify low-level mtDNA methylation. It is a highly sensitive approach for quantifying single nucleosides following DNA digestion, distinguishing methylated and unmethylated cytosines based on their mass and charge ratio. In mouse brain samples, Dou et al. used LC-MS-detected globally low mtDNA methylation and hydroxymethylation levels [23]. Infantino and coworkers found lower global mtDNA methylation in Down Syndrome (DS) cells than in controls using LC-MS/MS, which correlated with reduced methyl donor S-adenosylmethionine (SAM) availability in mitochondria from DS cells [24]. With LC-MS/MS, Liu et al. observed that mtDNA hypermethylation is related to endothelial cell dysfunction in Alzheimer’s disease [25]. On the other hand, Matsuda and colleagues investigated mtDNA methylation in mouse liver using LC-MS [17]. Their analysis detected a methylation level of 0.3–0.5% in mtDNA, suggesting that the signal might be entirely or partly from nDNA contamination.

It is worth noting that mass spectrometry-based approaches require the rigorous separation of mtDNA from nDNA prior to DNA digestion. Although mtDNA copies per cell vary from tens to thousands [26], the human mitochondrial genome is approximately 400,000 times smaller than the nuclear genome (16.5 Kb vs. 3.1 × 2 Gb), and around 4% of cytosines are methylated in the human nuclear genome (approximately 80% of CpG positions) [27,28,29]. With this, considerable effort has been dedicated to excluding nDNA contamination during mtDNA isolation. For example, Matthew and collaborators compared five mtDNA isolation methods, including the Percoll gradient method and magnetic microbead method, which were considered efficient in separating mtDNA from nDNA [30]. However, they found that thoroughly removing nDNA contamination remained challenging. Recently, Zheng et al. developed an nDNA-excluding method for assessing mtDNA methylation by mass spectrometry [31]. Cells were lysed by an alkaline solution, after which nDNA was removed using RecBCD cutting, followed by restriction enzyme cutting. Finally, linearized mtDNA was isolated from the residual degraded nDNA via electrophoresis gel. With this method, the purified mtDNA was obtained from HEK 293T and HepG2 cells, but no 5 mdC signals were detected by LC-MS/MS. Interestingly, a low abundance of 5 mdC was detected in mtDNA from NIH-3T3 cells, a mouse fibroblast cell line. In a previous study, we compared isolating mtDNA using an mtDNA isolation kit and the TRIzol method, confirming that mtDNA molecules are partly distributed in the TRIzol RNA phase [32,33]. Importantly, qPCR detected no nDNA contamination in the TRIzol RNA phase, suggesting its promising potential for LC-MS/MS to quantify mtDNA methylation levels.

In this study, we isolated mtDNA from the TRIzol RNA phase to overcome nDNA contamination. We first examined, by pyrosequencing, whether methylated mtDNA was preferentially enriched in the TRIzol RNA or TRIzol DNA phase. Next, we measured, by LC-MS/MS, the mtDNA global methylation in a panel of HepG2 transgenic cell lines stably expressing mitochondrial-targeted DNA methyltransferases. Furthermore, since mtDNA is the only genetic material in platelets, we set out to study the mtDNA methylation levels in platelets by LC-MS/MS, utilizing the mtDNA isolated from TRIzol RNA and compared to the total DNA isolation. Finally, we assessed the effectiveness of the established method for mouse brain tissue and assessed whether the isolation of mitochondria prior to TRIzol RNA isolation can further improve the purification of mtDNA isolated from brain tissues.

## 2. Results

### 2.1. Isolation of mtDNA from the TRIzol RNA Phase to Mitigate nDNA Contamination

To evaluate the yield of the mtDNA isolated from the TRIzol RNA phase and assess the potential nDNA contamination, qPCR was performed for the TRIzol RNA, TRIzol DNA, and gDNA isolation. To prevent cell type-specific bias, the analysis was performed in three different human cell lines: HepG2, HEK293T, and MCF7. While in the TRIzol RNA isolated samples the Ct values for nDNA genes (*GAPDH* and *ACTB*) were undetected, average Ct values of 27.4 and 28.0 were obtained for the TRIzol DNA and gDNA isolations, respectively, reflecting the presence of nDNA in the latter two isolations, as expected. Importantly, the TRIzol RNA phase yielded a considerable amount of mtDNA for subsequent analysis (Ct values for D-loop were on average 17.1 in TRIzol RNA, while they were 17.8 and 16.0 in TRIzol DNA and gDNA, respectively), while nDNA contamination was not detected (Figure 1A). Similar results were obtained for the MCF7 and HEK293T cell lines (Figure 1B,C).

To assess the consistency of mtDNA yield in the TRIzol RNA phase, a circular plasmid DNA (HBV 1.3-mer P-null replicon, sequence absent in the HepG2 genome) was used as an internal control for the isolation methods to mimic circular mtDNA. qPCR showed consistent mtDNA content (relative to *GAPDH* DNA) in gDNA isolations (Appendix A, right panel), which is the isolation method generally used for mtDNA copy number determination. However, this consistent pattern was not observed in the TRIzol RNA (Appendix A, left panel) or TRIzol DNA phase (Appendix A, middle panel).

Together, these findings suggest that the mtDNA isolated from the TRIzol RNA phase holds significant potential for mtDNA methylation measurement, although it does not suit mtDNA copy number quantification. Moreover, the spike-in of HBV plasmid DNA supports our hypothesis that the distribution of mtDNA in the TRIzol RNA phase may be due to its small circular structure.

### 2.2. Methylated mtDNA Is Not Preferentially Enriched over Unmethylated mtDNA in the TRIzol RNA Phase (Or Vice Versa)

To use the mtDNA isolated from the TRIzol RNA phase for methylation quantification, we first examined whether methylated mtDNA was enriched over unmethylated mtDNA in either of the TRIzol phases and compared it to gDNA isolation. Pyrosequencing showed no differences in mtDNA methylation levels among the three isolation procedures for the three assessed mtDNA regions, including each CpN site methylation and the mean methylation of the region (Figure 2). These results were validated using MCF7 cells (Appendix A), confirming the reliability of using mtDNA from the TRIzol RNA phase for mtDNA methylation quantification.

### 2.3. Assessment of mtDNA Methylation by LC-MS/MS

We then assessed the utility of LC-MS/MS for DNA methylation measurement using TRIzol RNA isolated samples. Analytes that have the same *m*/*z* transitions were baseline separated, as shown in the representative chromatogram of the calibration curve (Appendix A). This indicates that the presence of RNA analytes in the TRIzol RNA phase was not interfering with the measurements of DNA analytes, nor did the DNA analytes interfere with one another. The linear regression analysis of calibration curves showed excellent linearity with the coefficient of determination (R^2^), consistently ≥ 0.99. A circular plasmid DNA (HBV 1.3-mer P-null replicon) in high and low concentrations was used for LC-MS/MS imprecision. The intra-assay imprecision (CV%) of all detected analytes reached the requirement (CV < 10%) (Appendix A). 5-Hydroxymethyl-2′-deoxycytidine was not detected due to the low abundance in this plasmid DNA. Inter-assay, the imprecision of detected analytes was <15%, except for two nucleosides: 5-Methyl-2′-deoxycytidine and cytidine (Appendix A). Due to the very low abundance of 5-Methylcytidine in the plasmid sample, the CV% in the low-concentration sample was 22.2%. Also, the CV% for cytidine in the low- and high-concentration samples was 18.6% and 18.4%, respectively, which can be attributed to the absence of isotope-labeled internal standards. The limit of detection (LOD) and the limit of quantification (LOQ) were sufficient to measure mtDNA methylation in biological samples (Appendix A).

The regression analysis for the DNA cytosine methylation series yielded a slope of 1.045 when plotting measured values on the Y-axis against expected values on the X-axis (Figure 3A). The 95% confidence interval of the slope is 1.04 to 1.05, and that of the Y intercept is 0.018 to 0.069. The slope suggests that the measured methylation is consistent with the expected methylation, demonstrating the validity of the measurements. We next tested the method’s applicability by analyzing three independent samples of three human cell lines: HepG2, MCF7, and HEK 293T. Consistent low DNA methylation levels (~1.0–2.5%) were found in these human cell lines isolated from the TRIzol RNA phase, compared to those from the TRIzol DNA and gDNA isolations (Figure 3B).

### 2.4. LC-MS/MS Analysis Confirmed the Increased mtDNA Methylation in HepG2 Transgenic Cell Lines

We previously generated HepG2 transgenic cell lines expressing mitochondrial-targeted DNA methyltransferases [10]. In these cell lines, the expected increase in the methylation level (CpG methylation in the HepG2-MSssI cell line, and GpC methylation in the HepG2-MCviPI cell line) was confirmed using nanopore sequencing. In the current study, we evaluated the global DNA methylation differences in these cell lines by LC-MS/MS. Cells from either (i) three different passages or (ii) the same passage (P15) cultured in three independent flasks were harvested for mtDNA methylation measurement. LC-MS/MS confirmed the higher mtDNA methylation levels in the MCviPI- (2.89 ± 0.58%, mean ± SD) and MSssI- (2.03 ± 0.16%) expressing cells when isolated from the TRIzol RNA phase (Figure 4A). No difference was detected between the two control cell lines (wild type (WT) and the MCviPI_mutant (1.36 ± 0.23% and 1.39 ± 0.48%)). Similar methylation patterns for the four HepG2 cell lines were also detected in the TRIzol RNA isolates from the same passage (P15) (Appendix A). As expected, the differential DNA methylation could not be detected for these cell lines where total DNA was isolated by TRIzol DNA or gDNA isolation procedures (Figure 4B,C), due to the presence of nDNA affecting the methylation measurement. Lastly, the ratio between the measured concentrations of 2 dC and dT in all HepG2 transgenic cell lines was calculated (0.74 for mtDNA isolated from the TRIzol RNA phase, 0.62 for TRIzol DNA, and 0.61 for gDNA) and was in agreement with the reported Homo sapiens’ sequence (0.80 for mtDNA [34] and 0.69 for nDNA [35] (Figure 4D). Taken together, these findings demonstrated the precision and utility of LC-MS/MS measurement in mtDNA methylation, using mtDNA isolated from the TRIzol RNA phase.

### 2.5. LC-MS/MS Demonstrated Lower Methylation Level in DNA Isolated from the TRIzol RNA Phase in Clinically Relevant Samples Compared to TRIzol DNA and gDNA Isolations

To assess whether the TRIzol RNA isolation procedure, followed by LC-MS/MS, could apply to mtDNA methylation quantification in clinically relevant samples, we measured the global mtDNA methylation levels in platelets obtained from human blood samples. Firstly, platelets were obtained from blood and divided into two equal portions; mtDNA was then isolated using the TRIzol RNA phase or gDNA isolation method (Figure 5A). LC-MS/MS analysis showed a methylation level of 4.30 ± 0.22% for total DNA isolated from platelets (Figure 5B). Meanwhile, a significantly lower methylation level of 2.49 ± 0.43% was detected in the TRIzol RNA isolated platelets (Figure 5B). This significantly lower percentage of methylation was further confirmed in the additional six TRIzol RNA isolated platelet samples, with a mean methylation level of 1.46 ± 0.52%, compared to 4.34 ± 0.50% detected in thirty-eight additional platelet gDNA isolates (Figure 5C).

Despite lacking nuclear DNA, platelet isolation could be contaminated with nucleated cells, such as lymphocytes. Indeed, qPCR demonstrated significant nDNA contamination in the total DNA isolated from platelets, with mean Ct values of 26.4 and 26.0 for *GAPDH* and *ACTB*, respectively. In comparison, nDNA was barely detected in the TRIzol RNA isolates, showing a mean Ct value of 38.0 for both *GAPDH* and *ACTB*. Notably, mtDNA in the TRIzol RNA phase exhibited a mean Ct value of 15.9, which is only around 2 Ct higher than that of gDNA isolates, suggesting a substantial recovery of mtDNA from the TRIzol RNA phase (Figure 5D). These results implied a significant reduction in the overestimation of mtDNA methylation measurements due to nDNA contamination, thereby further improving the accuracy of LC-MS/MS for mtDNA methylation in clinical measurements.

### 2.6. LC-MS/MS Confirmed the Low Methylation of mtDNA Isolated by TRIzol RNA Isolation for Mouse Brain Tissue

To evaluate the effectiveness of the TRIzol RNA method for assessing mtDNA methylation in brain tissue using LC-MS/MS, DNA was isolated from mouse brain using two approaches (Figure 6A). Briefly, each mouse brain was divided into two halves. One half was directly homogenized and divided into eight tubes to prevent excessive DNA yield. One-eighth of this homogenized half brain was used for mtDNA isolation from the TRIzol RNA phase, while another eighth was used for gDNA isolation. The other half of the brain was processed to obtain the crude mitochondrial fraction (Crude-mt), which was then evenly divided and used for mtDNA isolation by the TRIzol RNA or gDNA method. For both homogenized brain tissues and the Crude-mt preparations, nDNA contamination (*Gapdh*) was detected in the gDNA isolates, but not in the TRIzol RNA isolates (Figure 6B lower panel). Importantly, a considerable amount of mtDNA content (*mt-Co2*) was obtained from the TRIzol RNA phase for both brain tissue (average Ct: 24.8) and the Crude-mt (average Ct: 21.2) (Figure 6B upper panel).

Finally, LC-MS/MS analysis showed that, in both homogenized brain tissue and Crude-mt, TRIzol RNA isolation demonstrated significantly lower DNA methylation levels (1.79 ± 0.98% and 2.5 ± 0.47%) compared with those in the gDNA isolates (5.12 ± 0.39% and 4.56 ± 0.34%) (Figure 6C). Interestingly, isolating the Crude-mt from mouse brain prior to mtDNA isolation from the TRIzol RNA phase provided no further substantial reduction in nDNA contamination over direct mtDNA isolation from brain tissue. The higher methylation level in Crude-mt isolated by the TRIzol RNA phase may indicate increased nDNA contamination, potentially resulting from the lower total DNA yield from Crude-mt, consequently reducing mtDNA representation. Collectively, we demonstrated that the LC-MS/MS measurement of mtDNA methylation in brain tissue benefits from using mtDNA isolated from the TRIzol RNA phase.

## 3. Discussion

The existence and functional significance of mtDNA methylation remain heavily debated. In our previous exploratory study, we confirmed, in the TRIzol RNA phase, the presence of mtDNA and detected no nDNA contamination [33]. In this study, we further developed the LC-MS/MS-based method for mtDNA global methylation quantification, utilizing the mtDNA isolated from the TRIzol RNA phase. We showed, by pyrosequencing, that the methylated mtDNA was not preferentially enriched over unmethylated mtDNA in the TRIzol RNA or TRIzol DNA phase. We then demonstrated that LC-MS/MS distinguished clearly between DNA nucleotides (2′deoxycytidine and 5-methyl-2′deoxycytidine) and RNA nucleotides (cytidine and 5-methylcytidine) because of their distinct molecular weights and chemical structures. This allowed us to rule out the influence of RNA on the mtDNA methylation assessment by LC-MS/MS. In a panel of differential mtDNA methylation engineered HepG2 cell lines, the LC-MS/MS method effectively detected the expected enhanced mtDNA methylation. Furthermore, our LC-MS/MS analysis was applied to various human cell lines and clinically relevant materials, including platelets and mouse brain tissues. When assessing mtDNA methylation using mtDNA from the TRIzol RNA phase, we observed consistently lower mtDNA methylation levels compared to those from total DNA isolates, demonstrating nDNA reduction. Overall, isolating mtDNA from the TRIzol RNA phase followed by LC-MS/MS analysis provides a time- and cost-effective approach for assessing mtDNA global methylation. Additionally, the TRIzol RNA isolation enabled us to investigate mtDNA gene expression by qRT-PCR, as well as mtDNA methylation by pyrosequencing and LC-MS/MS in one isolation.

In our previous study, a panel of HepG2 transgenic cell lines was constructed to express mitochondria-targeted cytosine DNA methyltransferases, which are expected to increase mtDNA methylation [10]. Indeed, Mposhi et al. demonstrated, using nanopore sequencing, the enhanced mtDNA GpC and CpG methylation level in HepG2 MCviPI- and MSssI-expressing cell lines compared with control cell lines (HepG2 WT and MCviPI_mut). In line with this finding, using the same transgenic cells isolated with TRIzol RNA, we confirmed, via LC-MS/MS, the increased global mtDNA methylation in HepG2 MCviPI- and MSssI-expressing cell lines compared to that in control cell lines (WT and MCviPI_mutant). Theys and coworkers also confirmed, using nanopore sequencing, that a 20% increase in methylation frequency was observed in both MCviPI and MSssI cell lines compared to WT and mutant controls [11]. Notably, their study revealed a higher specificity of MSssI inducing CpG methylation, with rarely detected GpC methylation; meanwhile, MCviPI induced 10% CpG methylation in addition to GpC methylation. This aligned with our observation of higher global methylation in the MCviPI-expressing cell line than that in the MSssI cell line. Together, we demonstrated the effectiveness of measuring mtDNA methylation using mtDNA isolated from the TRIzol RNA phase for LC-MS/MS.

In the analysis for mtDNA methylation in platelets, LC-MS/MS showed a methylation level at 4.30 ± 0.22%, which is in line with the reported genomic DNA methylation levels [27,28,29]. Meanwhile, qPCR showed that the gDNA isolates from platelets indeed contained nDNA contamination, which may be due to the contamination of leukocytes during platelet isolation [36,37]. This highlighted the importance of avoiding nDNA contamination for mtDNA methylation measurements, which was efficiently achieved in this study by the TRIzol RNA isolation. In the analysis of mtDNA methylation in mouse brain tissue, our results showed consistently lower global mtDNA methylation in TRIzol RNA isolations (1.79% on average) as measured by LC-MS/MS. This is in line with the genome-wide study of mtDNA methylation in the brain using bisulfite sequencing, reporting an average mtDNA methylation of 2.08 ± 0.98% in brain tissue [38]. In all, we showed the effectiveness of combining TRIzol RNA and LC-MS/MS for the quantification of mtDNA global methylation in human cell lines, platelets, and mouse brain tissues.

In this study, nDNA contamination in the TRIzol RNA phase was not detected by qPCR. It is worth noting that, while the significantly lower methylation level was reproducibly obtained from the TRIzol RNA phase across all measurements, 1 ng of nucleic acids was taken and examined by qPCR, yet 2500 ng was used for LC-MS/MS analysis. Therefore, the chances of including a trace of nDNA contamination in the analysis may have increased to interfere with the detected mtDNA methylation signal. On the other hand, the degree of the nDNA contamination and the mtDNA yield in the TRIzol RNA phase may vary across experiments, influenced by factors such as sample quality, cell numbers, and different operators. Considering the approximately 400,000-fold size difference between mtDNA (16.5 Kb) and nDNA (3.1 × 2 Gb) in the human genome, even trace amount of nDNA contamination could significantly influence the mtDNA methylation readouts. Also, around 4% cytosines and 80% CpG sites in nDNA are methylated [27,28,29], while approximately 2% global methylation was detected in mtDNA [38]. These factors hampered us from concluding the existence of mtDNA methylation in the samples in this work. Thus, the further purification of mtDNA needs to be considered for LC-MS/MS analysis, such as using plasmid-safe ATP-dependent DNase [39,40] to eliminate the potential presence of linear nDNA in the TRIzol RNA phase, followed by excising the mtDNA-specific band from electrophoresis gel [31].

Clinical investigations into mtDNA methylation grow because of its implications for disease pathogenesis [8]. In metabolic disorders, Mposhi’s and Theys’s studies, which used HepG2 transgenic cell lines, both revealed that an artificial induction of mtDNA methylation was related to changes in gene expression and metabolic functions [10,11]. In neurodegenerative diseases, where mitochondrial dysfunction plays an important role, increasing evidence suggests its potential links with mtDNA methylation [9,41]. On the other hand, while the cytosine methylation in mtDNA is under debate, studies on N6-methyladenosine (6 mA) appear to yield more consistent and reproducible findings. 6 mA is a novel methylation marker significantly enriched in mammals’ mtDNA, compared to nDNA [40,42]. 6mA methylation, although very low (no higher than 0.05%, measured by UHPLC-MS/MS or ELISA), has been shown to play regulatory roles, mediated by methyltransferase-like protein 4 (METTL4) [40,43]. Using mass spectrometry-based methods, 6mA methylation has been found to increase in mtDNA during ageing across species, including *Caenorhabditis elegans*, *Drosophila melanogaster,* and dogs [44,45]. This suggested an evolutionarily conserved mechanism, supporting 6 mA as a reliable marker of aging in addition to cytosine methylation. It is worth noting that the LC-MS/MS method developed in our study is applicable for quantifying 6mA methylation, enabling the concurrent investigation of both mtDNA methylation markers in the same sample at once in future studies.

## 4. Material and Methods

### 4.1. Cell Culture and Mouse Brain Tissues

Human hepatocellular carcinoma cells, HepG2 (ATCC: HB-8065, Manassas, VA, USA), human estrogen receptor positive breast cancer cells, MCF-7 (ATCC: HTB-22, Manassas, VA, USA), and human embryonic kidney cells, HEK293T (ATCC: CRL-3216, Manassas, VA, USA), were cultured in DMEM (Lonza BioWhittaker #BE12-604F, Basel, Switzerland), supplemented with 10% fetal bovine serum (FBS), 2 mM L-glutamine, and 50 μg/mL gentamycin sulfate. Cells were cultured in a humidified incubator with 5% CO_2_ at 37 °C.

HepG2 transgenic cell lines were constructed and cultured as described [10]. Briefly, HepG2 cells were engineered to stably produce mitochondria-targeted viral and prokaryotic cytosine DNA methyltransferases (mtM.CviPI or mtM.SssI for GpC or CpG methylation). A catalytically inactive variant (mtM.CviPI-Mutant) was created as a negative control. Antibiotic selection was performed on three HepG2 transgenic cell lines, with 1 µg/mL puromycin.

Four C57BL/6 J male mice from the central animal laboratory at the University of Groningen were housed and handled in accordance with Dutch standards and guidelines (Protocol 171224-01-003). All experiments were approved by the University of Groningen Committee for Animal Experimentation (CCD) of the University of Groningen (CCD license number AVD10500202215899; date: 1 September 2022).

### 4.2. Isolation of mtDNA and Total Genomic DNA

For all comparisons among TRIzol RNA, TRIzol DNA, and gDNA isolates, a pool of cells was first split into two equal portions, then isolated by the TRIzol reagent (Thermo Scientific, Waltham, MA, USA) or conventional phenol-chloroform isolation (gDNA isolation). TRIzol isolation was performed as described in the manufacturer’s protocol. Final volumes were 20 µL for each isolation.

For gDNA isolation, frozen cell pellets were resuspended in 1 mL 1× SE solution (NaCl, EDTA) and divided into two clean 1.5 mL Eppendorf tubes, 500 µL each. Then, 5 µL RNase A was added to each tube and incubated for 30–60 min at 37 °C. Afterwards, 5 µL proteinase K and 50 µL 10% SDS were added to each tube to remove proteins. After overnight incubation in a 55 °C water bath, 0.4 volume 6 M NaCl and 1 volume chloroform were added, and then rotated in a top-over-top rotator for 30 min. After centrifugation at 2500× *g* for 10 min at 20 °C, two layers were formed. The supernatant was taken into a clean 1.5 mL Eppendorf, and the same volume of isopropanol was added to precipitate DNA, then mixed gently until white threads of DNA formed a visible clump. The samples were centrifuged at 13,000× *g* for 15 min at 4 °C, and the supernatant was carefully removed without disturbing the DNA pellet. The pellet was washed twice by adding 500 µL of 70% ethanol. DNA pellets were eluted with 20 µL MilliQ H_2_O and concentration was determined by a NanoPhotometer™ (Implen GmbH, Munich, Germany).

### 4.3. Quantitative Real-Time PCR

The mtDNA relative copy number was determined by quantitative real-time PCR, using SYBR™ Green PCR Master Mix (Thermo Scientific, Waltham, MA, USA); 1 ng input was applied to keep Ct values within a proper range (15–30 Ct). The real-time qPCR was run for 40 cycles.

In human samples, to assess the mtDNA relative copy number, mtDNA (D-loop) was normalized with *GAPDH* DNA in the TRIzol DNA phase and gDNA isolates, and with *GAPDH* cDNA in the TRIzol RNA phase. *MT-CYB* (mtDNA) and *ACTB* (nDNA) were used for assessing mtDNA and nDNA content.

In mouse brain tissue, *mt-Co2* (mtDNA) and *Gapdh* (nDNA) were employed to assess mtDNA and nDNA content.

HBV 1.3-mer P-null replicon (6821 bp) was a gift from Wang-Shick Ryu (Addgene plasmid # 65462; https://www.addgene.org/65462/, accessed on 5 September 2025; RRID: Addgene 65462). As the first step of the isolation, 50 ng plasmid DNA was added into the TRIzol and gDNA isolation solution.

All primers are listed in Appendix A.

### 4.4. Pyrosequencing

MtDNA linearization was performed prior to bisulfite conversion by FastDigest HindIII (Thermo Scientific), according to the manufacturer’s protocol. Restriction cutting sites on human mtDNA have been checked by SnapGene^®^ software, version 8.1.1 (Dotmatics, Boston, MA, USA). In total, 500 ng TRIzol RNA isolates were used for bisulfite conversion by EZ DNA Methylation-Gold Kit (Zymo Research, Irvine, CA, USA), followed by Pyro-PCR to amplify human mtDNA D-loop, *MT-CYB*, and *MT-CO2* regions, using a pyrosequencing PCR kit (Qiagen, Hilden, Germany). Primers were designed using the PyroMark Assay Design 2.0 software (Qiagen, Hilden, Germany); a BLAST search (https://blast.ncbi.nlm.nih.gov/Blast.cgi, accessed on 5 September 2025) was carried out to exclude primers recognizing nuclear mitochondrial DNA sequences (NUMTs). PCR product integrity was checked by 1% agarose gel electrophoresis. The methylation percentage at each cytosine in any context (CpN) site was quantitatively analyzed using an AQ assay (to include non-CpG positions) in PyroMark Q24 Autoprep Software, version 2.0.8 (Qiagen, Hilden, Germany), according to the manufacturer’s guidelines.

All pyrosequencing primers are listed in Appendix A. The Heavy strand (H-strand) of mtDNA was targeted; targeted mtDNA sequences and targeted cytosines are listed in Appendix A.

### 4.5. Isolation of Platelets from Blood Samples

The method used in this study was adapted from the method described by van Doormaal et al. in 1988 [46] (Figure 5A). Leftovers from EDTA blood samples of the Lifelines cohort were centrifuged for 15 min at 200× *g* and 4 °C. The platelet-rich plasma (PRP) fraction was collected and placed on ice during the collection procedure. This PRP fraction was again centrifuged for 10 min at 200× *g* and 4 °C to get rid of the leukocytes. The PRP fraction was collected, and the samples were then centrifuged for 10 min at 2000× *g* and 4 °C to obtain a pellet of platelets. The pellet was then washed with 0.9% NaCl and centrifuged for 10 min at 2000× *g* and 4 °C. Next, the pellet was carefully suspended in 0.9% NaCl. Of this suspension, the platelet, white blood cell, and red blood cell content was assessed using an XN-10 Automated Haematology Analyzer (Sysmex, Kobe, Hyogo, Japan). The platelet suspension was then centrifuged again for 10 min at 2000× *g* and 4 °C, after which the platelet pellet was stored in the freezer at −20 °C.

All participants provided written consent. The Lifelines Cohort Study was conducted according to the principles of the Declaration of Helsinki and approved by the Medical Ethical Committee of the University Medical Center Groningen, The Netherlands.

### 4.6. Crude Mitochondrial Fraction Isolation

This method was adapted from Wieckowski and collaborators [47]. Brains were removed from adult C57Bl/6 mice. Half of the brain was used to isolate the crude mitochondrial fraction (Figure 6A). Tissue homogenization began with the disruption of fresh tissue in 1 mL of mitochondrial isolation buffer, freshly prepared with 225 mM mannitol (M4125), 75 mM sucrose (S9378), 30 mM Tris–HCl (CAS 1185-53-1), and 0.5 mM of EDTA (#6381-92-6) pH 7.4. The tissue was passed 5 times through 1.0 mL single-use syringes (B. Braun, Omnifix^®^-FLuer Solo, #9161406V, Melsungen, Germany) and needles of 0.90 × 40 mm diameter (B. Braun, 100 Sterican^®^, #4657519, Melsungen, Germany). The disrupted tissue was filtered through a 100 μm sieve (Corning Inc., Corning, NY, USA, 431752). The filtered cell suspension was homogenized using a semi-automatic pump-controlled cell rupture system, which is equipped with two syringes (SGE, Trajan© Scientific, Ringwood, Australia), a cell homogenizer (Isobiotech, EMBL, Heidelberg, Germany), a 10 μm bead, and a pump (ProSense B.V, Oosterhout, The Netherlands). First, the cell homogenizer was washed three times with a mitochondrial isolation buffer, and the cell suspension was pumped through the device. The resultant cell lysate was collected in a 1.5 mL Eppendorf, and after each sample the cell homogenizer was washed three times. The conventional centrifugation involved a slow spin at 600× *g* at 4 °C for 5 min to remove unbroken cells and nuclei. The supernatant containing the mitochondrial fraction was centrifuged at 10,000× *g* at 4 °C for 10 min. The supernatant containing the cytosol was removed and the crude mitochondrial pellet was stored at −20 °C. This pellet was used for further analysis. To enhance mitochondrial integrity, the whole process must be carried out at 4 °C throughout; hence, all buffers, reagents, and samples were stored on ice during the experiments.

The TRIzol RNA isolation and gDNA isolation of the mouse brain or the Crude-mt fraction followed the same procedure as described above.

### 4.7. LC-MS/MS

The acetonitrile, ammonium acetate, and acetic acid were of analytical grade (Merck, Darmstadt, Germany). Thymidine, 2′-deoxycytidine, cytidine, and 5-methylcytidine were purchased from Sigma-Aldrich (Sigma-Aldrich, St. Louis, MO, USA). 2′-deoxycytidine-15N3 and Thymidine-13C10, 15N2 were purchased from Cambridge Isotope Laboratories (Tewksbury, MA, USA). 5-methyl-2′-deoxycytidine-d3,5-(hydroxy) methyl-2′-deoxycytidine, and 5-(hydroxy) methyl-2′-deoxycytidine-d3 were purchased from Bio-Connect (Bio-Connect, Huissen, The Netherlands). 5-methyl-2′-deoxycytidine was purchased from Thermo Scientific (USA). Liquid chromatography was performed on an ACQUITY UPLC BEH Amide 2.1 mm × 100 mm, 1.7 µm column (Waters, Milford, MA, USA).

Individual stock solutions were prepared by dissolving them to 1 mg/mL in distilled water. A working standard solution of 10 µg/mL for each analyte was prepared by diluting the intermediate stock standard solution. Nine calibrators were prepared by adding different volumes of the analyte working solution, with final concentration ranges of 0–100 ng/mL for 2′-deoxycytidine, thymidine, cytidine, and 5-methylcytidine, and of 0.2–10 ng/mL for 5-methyl-2′-deoxycytidine and 5-(hydroxy) methyl-2′-deoxycytidine. The internal standard concentrations were 2′-deoxycytidine-15N3 (1 ng/µL), 5-methyl-2′-deoxycytidine-d3 (0.1 ng/µL), and thymidine-13C10, 15N2 (2.5 µg/µL).

To ensure that the presence of RNA in the TRIzol RNA phase did not interfere with the mtDNA methylation analysis, two RNA nucleosides (cytidine and 5-methylcytidine (m5C)) were included together with the four DNA nucleosides that were used in the calibration curve of LC-MS/MS. Isotope-labelled internal standards were not used for these two RNA nucleosides, as they were only used to ensure baseline separation from other DNA nucleosides.

DNA or RNA was digested into single nucleosides after isolation (Nucleoside Digestion Mix, cat.no. M0649S, New England Biolabs, Ipswich, UK). The digestion reaction was prepared according to the manual protocol. For all TRIzol RNA isolated samples, 2500 ng was used as the input for LC-MS/MS, whereas 100 ng was used as the input for samples isolated by TRIzol DNA and gDNA isolation (because of the large size of nDNA molecules).

The mobile phase included A (0.8% acetic acid and 10 mmol/L ammonium acetate in aqueous solution) and B (0.1% acetic acid in acetonitrile) [48]. Additionally, a 5 µL sample solution was subjected to LC-MS/MS analysis. A Shimadzu LC-30AB U(H)PLC was used for a gradient elution, the conditions for which were 0–3 min isocratic 80% B-20% A; 3–4 min linear 80% B to 50% B; 4–4.1 min back to 80% B; and 7 min end of run. The flow rate was 0.3 mL/min. Mass spectrometry detection in the MRM mode was performed by using a SCIEX API 4500 (AB SCIEX, Framingham, MA, USA) equipped with an electrospray ionization (ESI) source in positive ionization mode. The desolvation gas flow rate was set to 50 L/h at a temperature of 500 °C, and the curtain gas flow rate was set to 30 L/h. The capillary voltage was set to 5500 V; the declustering potential (DP) and collision energy (CE) were dependent upon the *m*/*z* transition for each compound (Appendix A). For data acquisition, Analyst version 1.6.3 was used. Methylation percentage was determined as (5 mdC/(5 mdC + 5 hmC + 2 dC)) × 100.

The LC-MS/MS assay was validated by evaluating imprecision, limit of detection, limit of quantification, and linearity. Quality controls were prepared with the HBV plasmid DNA (HBV 1.3-mer P-null replicon) in high (0.2 µg/µL) and low (0.05 µg/µL) concentrations. Intra-assay was assessed by measuring the quality control samples on the same day, n = 20. Inter-assay imprecision was assessed by measuring the quality control sample on 6 different days. The LOD and the LOQ for each analyte of interest were calculated based on the standard deviation of the response and the slope of the calibration curve measured on three different days. Linearity was assessed by measuring the nine calibrators with the concentration range mentioned above.

The DNA cytosine methylation dilution series were prepared with a constant amount of 2′-deoxycytidine (2 dC) and gradually increasing amounts of 5-methyl-2′-deoxycytidine (5 mdC), yielding methylation levels ranging from 0.032% to 7.7%.

### 4.8. Statistical Analysis

Statistical analysis was performed using GraphPad Prism 10 software (San Diego, CA, USA). A paired *t* test was used to compare DNA methylation levels among different isolation methods, unless addressed differently.

## 5. Conclusions

Overall, our findings demonstrate that the overestimation of mtDNA global methylation in LC-MS/MS analysis resulting from nDNA contamination could be efficiently reduced by isolating mtDNA from the TRIzol RNA phase. Meanwhile, TRIzol RNA isolation also allowed us to assess, in one preparation, mtDNA gene expression by qRT-PCR and region-specific mtDNA methylation by pyrosequencing. With the developed method, we successfully quantified mtDNA global methylation in transgenic cell lines (artificially induced) and clinically relevant tissue samples. Overall, our method, validated in human platelets and mouse brains, provided a technical foundation for future investigations of mtDNA methylation alterations in neurometabolic disorders and other clinically relevant diseases.

## Figures and Tables

**Figure 1 ijms-26-08864-f001:**
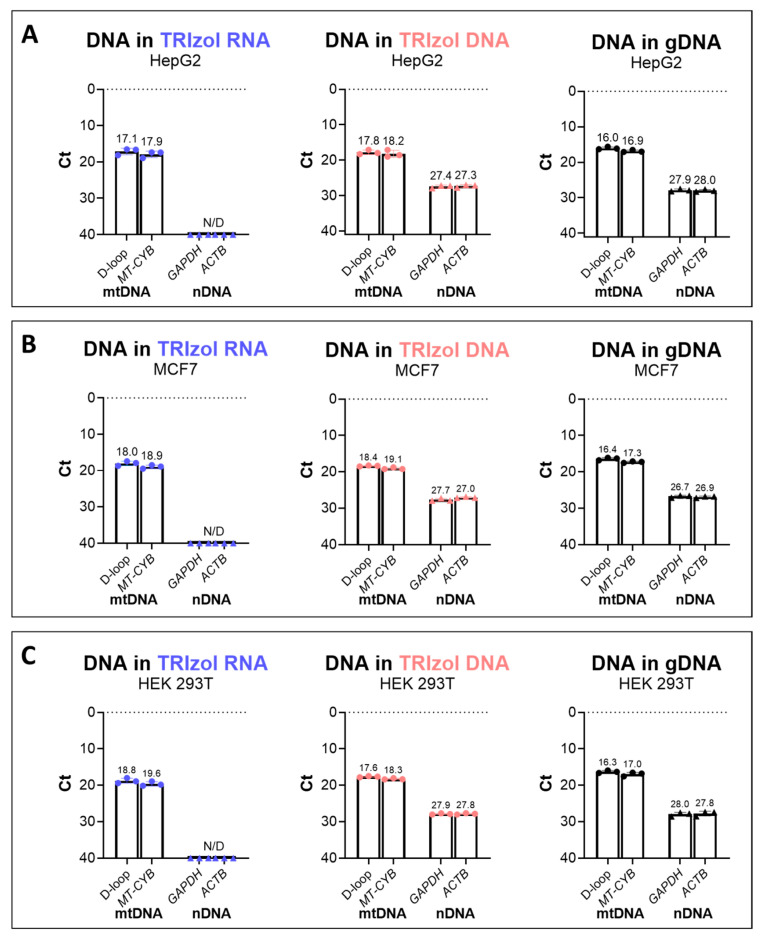
Relative yield of DNA in the TRIzol RNA, TRIzol DNA, and gDNA isolation of (**A**) HepG2, (**B**) MCF7, and (**C**) HEK 293T cell lines. DNA yield was measured by qPCR after isolation; average Ct values of mtDNA regions (D-loop and *MT-CYB*) and nDNA genes (*ACTB* and *GAPDH*) are shown for TRIzol RNA (shown in light blue in the left panels), TRIzol DNA (shown in pink in the middle panels), and gDNA (shown in black in the right panels) isolates, respectively (n = 3 for each cell line). MtDNA genes were represented as dots, nDNA genes were represented as triangles.

**Figure 2 ijms-26-08864-f002:**
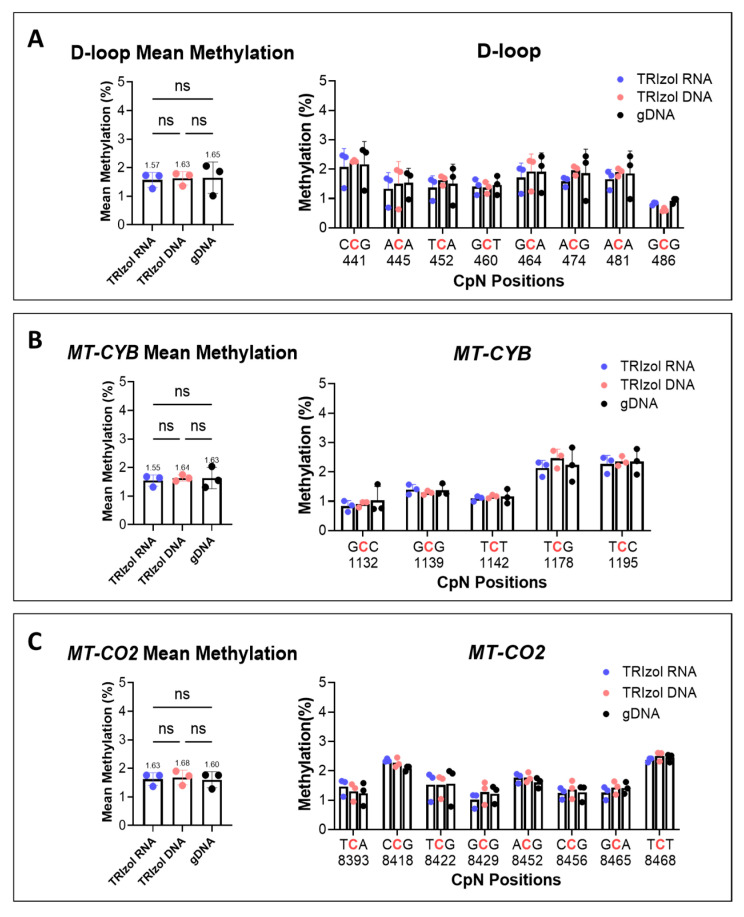
HepG2 mtDNA methylation measured by pyrosequencing. Cells were divided into two equal portions for TRIzol isolation and gDNA isolation. The methylation percentage was measured by PyroMark Q24 for three mtDNA regions, (**A**) D-loop, (**B**) *MT-CYB,* and (**C**) *MT-CO2*, using mtDNA from TRIzol RNA, TRIzol DNA, and gDNA isolations. The H-strand was targeted. Mean methylation was calculated as the mean of all CpN positions in the targeted regions. Statistical significance was determined using a paired *t* test, ns: no significant difference (n = 3).

**Figure 3 ijms-26-08864-f003:**
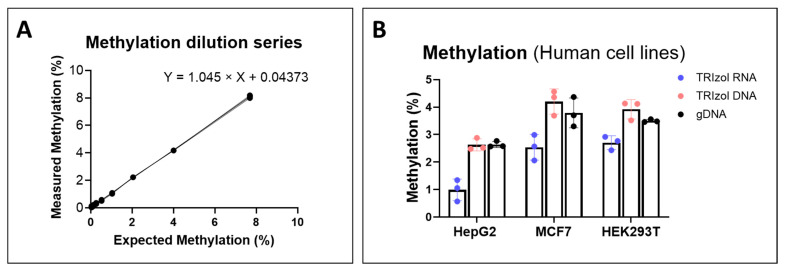
LC-MS/MS analytes and DNA cytosine methylation dilution series. (**A**) 2 dC methylation dilution curve. The 2 dC methylation dilution curve was prepared with a constant amount of 2′-deoxycytidine (2 dC) and gradually increasing amounts of 5-methyl-2′-deoxycytidine (5 mdC). (**B**) DNA methylation of HepG2, HEK293T, and MCF7 cell lines that were isolated by TRIzol RNA, TRIzol DNA, and gDNA isolations. n = 3 for both panels.

**Figure 4 ijms-26-08864-f004:**
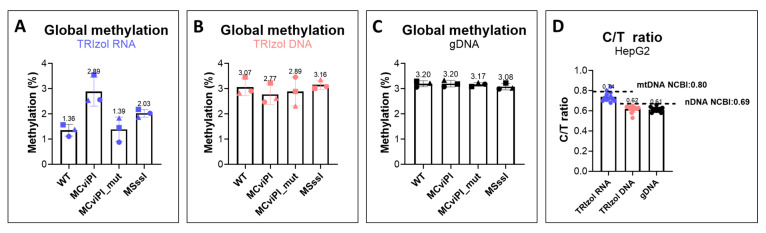
DNA methylation in HepG2 transgenic cell lines. DNA was isolated by the (**A**) TRIzol RNA, (**B**) TRIzol DNA, and (**C**) gDNA isolation procedures for cell lines from three different passages. The three different symbols represent the three different passages. (**D**) Ratio between the measured concentrations of total 2 dC and dT in HepG2 transgenic cell line samples. DNA methylation was measured by LC-MS/MS. WT represents the HepG2 wild-type cell line, MCviPI represents HepG2 cell transduced to stably express the MCviPI methyltransferases inducing GpC methylation, MCviPI_mut represents HepG2 cells transduced to stably express the mutant MCviPI methyltransferases, and MSssI represents HepG2 cells transduced to express the MSssI methyltransferases inducing CpG methylation (n = 3).

**Figure 5 ijms-26-08864-f005:**
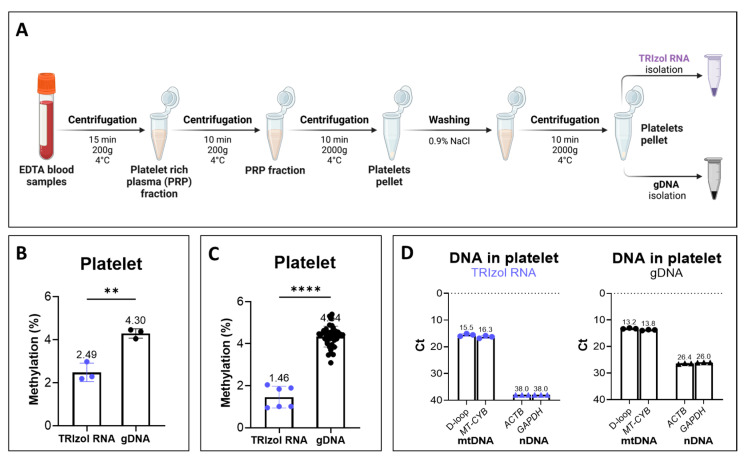
DNA global methylation level of platelets. Global DNA methylation was measured by LC-MS/MS. (**A**) Procedure for DNA isolation from platelets in human blood. Platelet pellets were divided into two equal portions and isolated by TRIzol RNA and gDNA. Figure created with BioRender.com. (**B**) LC-MS/MS analysis on platelet DNA methylation. Platelets were obtained from human blood and isolated by TRIzol RNA or gDNA. (**C**) DNA methylation measured by LC-MS/MS for an additional six platelet samples with DNA isolated by TRIzol RNA, and an additional thirty-eight gDNA isolated samples. (**D**) Relative yield of mtDNA and the nDNA contamination in TRIzol RNA and gDNA isolates. Ct values of mtDNA regions (D-loop and *MT-CYB*) and nDNA genes (*ACTB* and *GAPDH*) are shown for TRIzol RNA and gDNA isolates, respectively. MtDNA genes were represented as dots, nDNA genes were represented as triangles. Statistical significance was determined using the paired *t* test for platelets (n = 3, Figure 5B), and the Mann–Whitney test was used for additional platelet samples (Figure 5C); ** *p* < 0.01, **** *p* < 0.0001.

**Figure 6 ijms-26-08864-f006:**
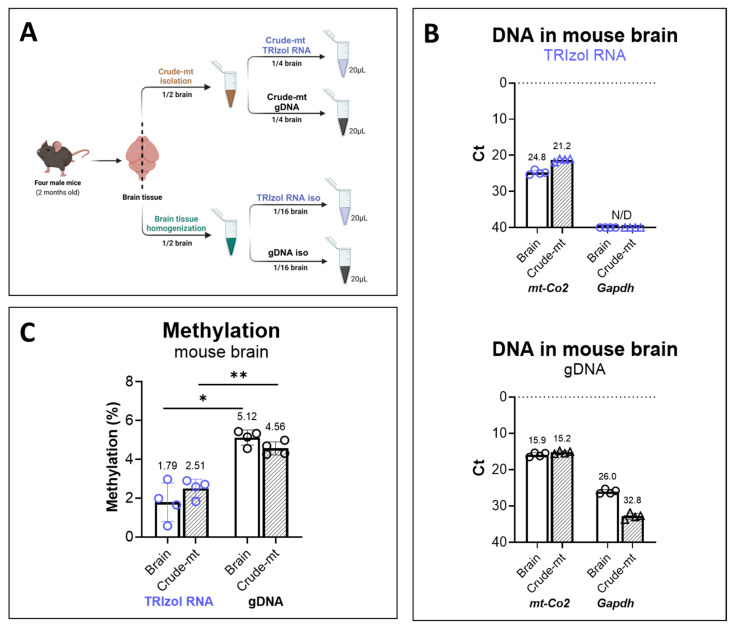
Assessment of mtDNA methylation of mouse brain tissue and nDNA contamination among different isolation methods. (**A**) Procedure of isolating mtDNA from mouse brain tissue for LC-MS/MS analysis. Brain tissues from two-month-old C57BL/6 male mice were collected and split into two parts. The first half (on the right) was utilized to extract a crude mitochondrial fraction (Crude-mt), followed by mtDNA isolation from the TRIzol RNA phase or genomic DNA isolation (Crude-mt gDNA) methods. The second half (left) was homogenized (and split into eight clean tubes) and utilized to isolate mtDNA from the TRIzol RNA phase or gDNA isolation (n = 4). Figure created with BioRender.com. (**B**) qPCR measurement of the nDNA and mtDNA relative contents in the TRIzol RNA and gDNA isolations. Ct values for the mtDNA gene (*mt-Co2*) and nDNA gene (*Gapdh*) are shown respectively for mtDNA isolated by TRIzol and gDNA from Crude-mt (represented by hallow triangles) and mouse brain (represented by hallow dots) samples. Crude-mt represents crude mitochondrial fraction isolated from mouse brains by the centrifugation method (n = 4). (**C**) DNA methylation in the TRIzol and gDNA samples from Crude-mt and from the whole brain (n = 4), measured by LC-MS/MS. Statistical significance was determined using a paired *t* test; * *p* < 0.05, ** *p* < 0.01.

## Data Availability

Data is contained within the article and Appendix A.

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
