# Peer review of "Effective Reduction in Nuclear DNA Contamination Allows Sensitive Mitochondrial DNA Methylation Determination by LC-MS/MS"

_ijms, 2025, doi:10.3390/ijms26188864_

Round 1
Reviewer 1 Report
Comments and Suggestions for Authors
This is a very smart and interesting manuscript as the authors elegantly found that the existence of high quantity of mitochondrial DNA (mtDNA) in TRIOL RNA phase instead of nuclear DNA (nDNA). What is more, the authors ingeniously designed experiments to profile the methylation of mtDNA finally in clinical samples. Even though in the field of epigenetics, whether the existence of methylation in miDNA is full of controversy, the manuscript opens a good way. What is more, the writing, workload and the creativity (especially for the experimental design) are good enough. Therefore, the manuscript is recommended to be accepted by the International Journal of Molecular Sciences. Here are some concerns for the authors to improve the quality of the manuscript:
1) I am afraid that the authors submitted the manuscript in a hurry, the authors should revise the manuscript in the right format according to the request of the journal.
2) The two paragraphs of the Abstract section should be merged in sole one.
3) For the Introduction section, Figure 1 should be moved to SPI section or merged with other Figures from the manuscript. Maybe moving it to the SPI section is a better choice.
4) When citing references, the authors tend to cite the reference in the middle of the sentences. In fact, the authors just need to replace it at the end of the sentences for better reading.
5) Here is an interesting question: Since the low abundance methylation level in mtDNA and the dynamic regulation of DNA methylation exists during epigenetic regulation process, have the authors considered mentioning the dynamic regulation of mtDNA methylation to explain the controversy to the Discussion section?
Author Response
Thank you very much for taking the time to review this manuscript. Please find the detailed responses below and the corresponding revisions/corrections highlighted/in track changes in the re-submitted files.
Comment 1
I am afraid that the authors submitted the manuscript in a hurry, the authors should revise the manuscript in the right format according to the request of the journal.
The authors apologize for any inconvenience that this action may have caused. The manuscript has now been revised in the format according to the request of the journal.
Comment 2
The two paragraphs of the Abstract section should be merged in sole one.
We appreciate the reviewer for pointing this out, the two paragraphs in the Abstract section have now been merged into one.
Comment 3
For the Introduction section, Figure 1 should be moved to SPI section or merged with other Figures from the manuscript. Maybe moving it to the SPI section is a better choice.
We appreciate the reviewer’s suggestion. Figure 1 is moved to the supplementary as Suppl. Fig. 1. All the rest main figures and supplementary figures have been reordered, and all figure citations of main figures and supplementary figures have been corrected accordingly.
Comment 4
When citing references, the authors tend to cite the reference in the middle of the sentences. In fact, the authors just need to replace it at the end of the sentences for better reading.
We appreciate the reviewer’s suggestion. The citing in the middle of the sentences, in the case that the entire sentence is related to the same reference, has now been moved to the end of the sentence.
In the meantime, in cases where the reference is placed in the middle of the sentences, we aimed to indicate the specific part of the statement that was cited from that particular reference, while the rest of the sentence may relate to another reference. For instance, line 58-61 in the text “Increasing evidence indicates that mtDNA methylation is involved in mitochondrial dysfunction and associated with the pathogenesis of various diseases7,8 including neurodegenerative diseases9, metabolic disorders10,11 and cardiovascular diseases12,13.”, in this sentence, the examples of diseases were cited from different publications. We feel that this helps to clearly attribute the information to the proper citations, making it more convenient for readers to locate specific resources and prevent confusion.
Comment 5
Here is an interesting question: Since the low abundance methylation level in mtDNA and the dynamic regulation of DNA methylation exists during epigenetic regulation process, have the authors considered mentioning the dynamic regulation of mtDNA methylation to explain the controversy to the Discussion section?
We appreciate the reviewer’s interesting question, and we agree with the possibility of dynamic regulation of DNA methylation as a potential explanation to explain the current controversies regarding the existence of mtDNA methylation.
However, we believe that the main concerns on the existence debates lie in technical limitations of mtDNA methylation measurements. Therefore, the current work primarily focused on the development of the LC-MS/MS based method to quantify the methylation and using mtDNA isolated from the TRIzol RNA phase to reduce nDNA contamination.
The methylation measurement in this study used mtDNA from different human cell lines, human platelets and mouse brains. We do not find clear evidence for a dynamic regulation of mtDNA methylation, thus, we feel that extending the discussion of this concept could result in overstating the implications of our findings and lead to confusion among readers.
Based on the above, we chose to keep the discussion focused on the methodology’s importance and on the novelty of our work, while pointing out that the existence of mtDNA methylation requires further explorations. Investigating the dynamic changes of mtDNA would indeed benefit future studies with solid methylation quantification approach.
Reviewer 2 Report
Comments and Suggestions for Authors
I reviewed the article titled: “Effective reduction of nuclear DNA contamination allows sensitive mitochondrial DNA methylation determination by LC-MS/MS” and I found it very interesting, and very well-prepared. The authors described procedure of mtDNA isolation tDNA from the TRIzol RNA phase to overcome nDNA contamination in order to examine by pyrosequencing hether methylated mtDNA was preferentially enriched in the TRIzol RNA or
the TRIzol DNA phase. The introduction sufficiently presents the actual information regarding mtDNA methylation in mammals. The laboratory techniques are accurate; therefore, in my opinion, the results are reliable. The authors provided the vast supplementary material useful for other researchers to be able to replicate the experiments. Prior the publication please kindly check and correct the following points:
Remarks:
Figure 1 – According to HGNC database the protein-coding genes are marked with previous names (e. g. MTATP8 instead of MT-ATP8). In my opinion all gene names should be written in the same way on the figure. Moreover, not all genes are marked on the scheme, especially those encoded on the L strand. I would suggest to use one nomenclature in the whole manuscript.
Figure 3 – On X axis there are CpN positions for each gene/ region in human HepG2 cell line. D-loop positions from 441 to 486, MTCYB from 1132 to 1195 and MTCO2 from 8393 to 8468. However, the position of the MTCO2 in human reference genome is 7586-8269, whereas MTCYB is 14747-15887. It should be corrected in this figure and in supplementary material fig 2. as well on order to omit future mistakes in the interpretation.
Suppl. Table 3 – target location for MTCYB does not correspond to the positions of this gene in the genome. However, I checked the mtDNA sequence in BLAST and it shows a correct region ~15376-15439 corresponding with the position of human MTCYB gene in the reference genome. Please correct this information.
Author Response
Thank you very much for taking the time to review this manuscript. Please see the attachment with point-by-point response.

Round 2
Reviewer 1 Report
Comments and Suggestions for Authors
All the concerns have been revised or explained according to the 2nd edition of the manuscript. The present manuscript reads better than the last edition. Therefore, the present version is recommended to be accepted by the International Journal of Molecular Sciences. Congratulations to the authors!